# On the Reproducibility of Vision Transformers Need Registers

## Abstract

Training Vision Transformers (ViTs) presents significant challenges, one of which is the emergence of artifacts in attention maps, hindering their interpretability. Darcet et al. (2024) investigated this phenomenon and attributed it to the need of ViTs to store global information beyond the `[CLS]` token. They proposed a novel solution involving the addition of empty input tokens, named registers, which successfully eliminate artifacts and improve the clarity of attention maps. In this work, we reproduce the findings of Darcet et al. (2024) and evaluate the generalizability of their claims across multiple models, including DINO, DINOv2, OpenCLIP, and DeiT3. While we confirm the validity of several of their key claims, our results reveal that some claims do not extend universally to other models. Additionally, we explore the impact of model size, extending their findings to smaller models. Finally, we untie terminology inconsistencies found in the original paper and explain their impact when generalizing to a wider range of models.

## 1 Introduction

The pursuit of universal feature embeddings has driven advances in computer vision, with transformer-based architectures surpassing traditional methods (Vaswani et al., 2017). Pretrained transformers using large labeled datasets (e.g., DeiT3 on ImageNet-22K (Touvron et al., 2022)), text supervision (e.g., CLIP (Radford et al., 2021)), or self-supervised learning (e.g., MAE (He et al., 2021)) outperform older hand-crafted techniques like SIFT (Lowe, 2004). Fully learnable pipelines have popularized multiple transformer backbones, with the DINO framework (Caron et al., 2021) standing out. DINO produces high-quality, semantically meaningful features, with clear attention and feature maps that enable applications such as LOST (Siméoni et al., 2021), which improves object discovery accuracy by 15%. Similarly, TokenCut (Wang et al., 2022b) leverages ViT attention maps for unsupervised object segmentation. Building on DINO's success, DINOv2 (Oquab et al., 2023) refined data processing to curate a 142-million-image dataset, excelling in dense prediction tasks.

Darcet et al. (2024) investigated DINO and DINOv2, integrating LOST's feature extraction with additional input tokens. They found that, unlike DINO, DINOv2's attention and feature maps contain artifacts / high-norm tokens disrupting both object discovery, which is the main downstream task of LOST, as well as the interpretability of the results. These artifacts, prevalent in large models like DINOv2-G, DeiT3-L, and OpenCLIP-L, suggest that vision transformers repurpose redundant patches for internal computation and global information aggregation. While DINOv2 excels in benchmarks, these attention artifacts significantly reduce object discovery accuracy, aligning it with conventionally supervised models. In contrast, DINO remains artifact-free, making it an exception among modern transformers.

To mitigate this, Darcet et al. (2024) propose adding empty tokens named "registers" to the input, providing dedicated tokens for internal computation and eliminating artifacts. This idea is not new to the field, as dedicated memory tokens have previously been inserted into the transformer's input sequence to improve long-context machine translation and other NLP tasks (Burtsev & Sapunov, 2020). This solution not only resolves the issue but also marginally enhances model performance. Their findings highlight a common artifact issue in modern vision transformers and offer a practical method to improve interpretability in downstream tasks.

This paper investigates the claims of Darcet et al. (2024) by reproducing their results. We replicate all experiments involving DINOv2, while for DINO, DeiT3, and OpenCLIP, we focus on runs without registers due to the lack of available models. Additionally, we assess the generalizability of their results, as detailed in Section 2.

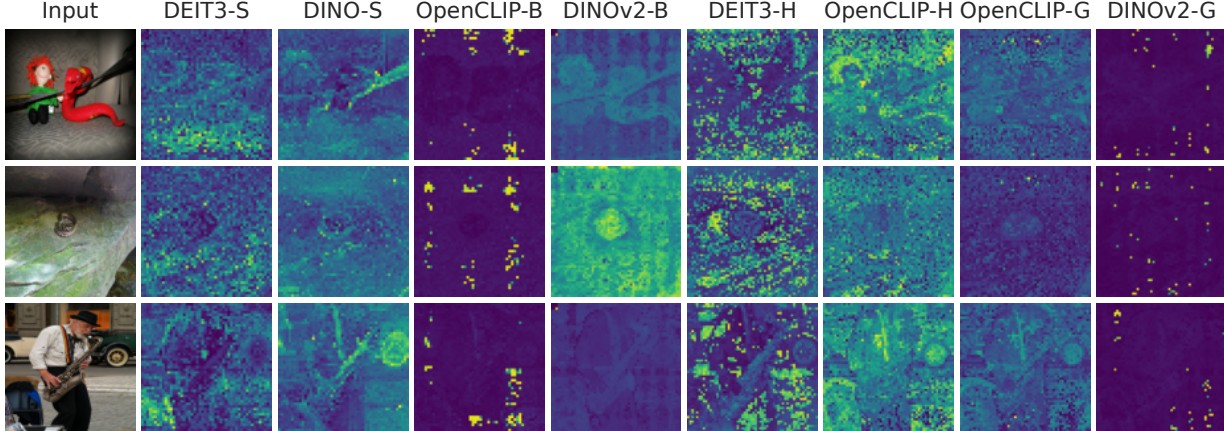

Figure 1: Feature maps generated by DeiT3, OpenCLIP, DINO, and DINOv2 models for three sample images, calculated in high resolution for better visualization. High-norm outliers are evident in most models except DINO-S and DINOv2-B, indicating that even smaller models struggle with outliers in their feature representations. Additionally, the absence of artifacts in DINO's feature maps supports the findings of Darcet et al. (2024). Appendix C includes a more in-depth analysis on the relation of tokens' L2 norm distribution and artifacts.

## 2  Scope of Reproducibility

This work verifies and extends the findings of Darcet et al. (2024), who investigate artifacts in vision models, their correlation with high-norm tokens, and propose registers— empty tokens added to the input — to address these issues. Below, we outline their claims.

**Claim 1: Artifacts are high-norm outlier tokens.** Tokens with exceptionally high L2 norm (top 2%) correspond to artifacts in attention and feature maps. The presence of such artifacts degrades the interpretability of both representations.

**Claim 2: Training large models (>300M parameters) leads to high-norm tokens.** As model size increases, high-norm tokens become more prominent, suggesting that scaling plays a key role in their emergence.

**Claim 3: High-norm tokens emerge in regions with redundant visual information.** Large vision transformers, seeking additional space for internal computations, repurpose tokens in image regions with highly redundant information without compromising performance. Consequently, high-norm tokens tend to appear in these regions.

**Claim 4: High-norm tokens hold little local information but encode global context.** Darcet et al. (2024) propose that high-norm tokens, repurposed for internal computation, lose local information while encoding global context. To investigate whether local information is preserved, the authors design two tasks: a position prediction task and a pixel reconstruction task. The position prediction task compares the ability of normal versus high-norm tokens to predict their grid position in the original image. The pixel reconstruction task evaluates how well normal and high-norm tokens reconstruct an image patch from their corresponding output embeddings. To assess whether high-norm tokens better capture global context, the authors employ an image classification task, using either a `[CLS]` token, a random normal token, or a random high-norm token as the final image representation.

**Claim 5: Adding registers eliminates artifacts in attention maps while maintaining or improving performance.** Introducing empty input tokens, referred to as registers, provides dedicated elements for internal computation. This prevents artifacts from appearing in feature maps and solves the aforementioned interpretability issues. Furthermore, the addition of registers does not degrade performance and occasionally improves it.

# 3 Methodology

In this section, we provide an overview of our experimental setup, encompassing models, datasets, training procedures, hardware, and our carbon footprint. All experiments and analyses are documented in our GitHub repository [1]. Relevant scripts are supplied to facilitate reproducibility and transparency. Where the original authors do not provide direct code, we offer our implementations. Below, we describe our setup in detail, noting any deviations from prior work and discussing the environmental impact of our computations.

## 3.1 Terminology

This section clarifies the definitions of "attention map," "feature map," "outlier," and "artifact," as these key concepts are central to our reproducibility study. In Darcet et al. (2024), their usage lacks clear distinctions, which could potentially lead to confusion. Here, we disentangle these terms to ensure a precise interpretation.

**Attention Maps** visualize attention scores between the `[CLS]` token and input tokens in an image-like grid. Following Darcet et al. (2024), we compute attention maps by averaging scores across attention heads in the final layer.

**Feature Maps** represent the L2 norm of output tokens in an image-like grid. As in Darcet et al. (2024), we calculate these using final-layer output tokens prior to the last Layer Normalization.

**High-norm / Outlier tokens** are output tokens with substantially higher norms than other tokens in the same model. The authors posit these typically occur in redundant patch regions, where they facilitate global information aggregation rather than local feature extraction. Both Darcet et al. (2024) and the present work analyze norms before the final block's Layer Normalization.

**Artifacts** are anomalous patterns in attention or feature maps. Feature map artifacts always correspond to high-norm tokens, while attention map artifacts manifest as disproportionately high `[CLS]` attention scores. The presence of artifacts reduces visualization interpretability, as the remaining patches have values too low to visualize clearly.

## 3.2 Description of Methods

To address the emergence of high-norm tokens, Darcet et al. (2024) introduce an approach that incorporates empty tokens—referred to as registers—into the vision transformer input stage. These additional tokens serve as dedicated computational units, reducing the model's reliance on redundant patches for internal computations and mitigating artifacts.

**Registers** act as placeholders in the token sequence, allowing vision transformers to compute internally without modifying image tokens or distorting feature representations. By preserving the integrity of meaningful tokens, they enhance feature map interpretability. Despite potential trade-offs, experiments show that models with registers maintain or even improve classification accuracy and downstream performance.

## 3.3 Experimental Setup

All implementations utilize the TIMM Python library [2] for model definitions and pretrained checkpoints. Unless otherwise stated, experiments are conducted on the ImageNet-1K (Deng et al., 2009) dataset, accessed via HuggingFace[3]. All experiments use a random seed of 42 to facilitate reproducibility.

---

[1] Official GitHub repository
[2] Timm Python Library
[3] HuggingFace Library

**Claims 1, 2.** We extract and visualize the `[CLS]` token's attention scores from the final ViT layer, averaging across all attention heads. For feature maps, we use the L2 norms of the model's output tokens before the final Layer Normalization. Both maps are reshaped into a grid and generated at two resolutions: *high* ($800 \times 800$) and *native* (the model-specific input resolution). We also randomly sample 5,000 images from the validation set, forward each image through the model, and plot the distribution of the L2 norms (before the final LayerNorm) of the output tokens [4]. For PVT (Wang et al., 2022a) and Swin (Liu et al., 2021) models—which lack a dedicated `[CLS]` token—we derive attention maps by averaging across both the query dimension and all attention heads, producing final attention scores that reflect how much focus each token receives from others, averaged over all heads.

**Claim 3.** Following the approach of Darcet et al. (2024) and using our 5,000-image subset, we compute the cosine similarity between each patch embedding and its four immediate neighbors after the patch embedding layer. From the distribution of L2 norms, we select a cutoff corresponding to the 98th percentile (top 2%), designating high-norm (outlier) tokens. We then plot two similarity distributions: one for these high-norm outlier tokens and one for the remaining tokens.

**Claims 4** Each model undergoes a two-stage process: (1) extracting output tokens, unnormalized norms (before the final LayerNorm), and `[CLS]` tokens for all training and validation images; (2) training task-specific classifiers on extracted tokens, evaluating performance by input category. For PVT (Wang et al., 2022a) and Swin (Liu et al., 2021) models, we employ average pooling over the spatial dimension—following their standard architectural implementations—to generate the image representations used by downstream classifiers. The pooled token will be referred to as `[CLS]` token. In position prediction and patch reconstruction, tokens are classified as normal or outlier, while image classification uses `[CLS]`, normal, and outlier tokens. Outliers are defined as tokens exceeding the 98th percentile of the L2-norm distribution - calculated on a per-model basis.

- **Position Prediction.** This task is formulated as classification, where each token predicts its original patch position. For a square image of size $(Y \times Y)$ with patch size $P$, the total patch positions are $\left(\frac{Y}{P}\right)^2$ - assuming $Y$ is evenly divisible by $P$. A linear layer maps the token embedding ($D$) to an integer in $[0, \left(\frac{Y}{P}\right)^2 - 1]$, which is then converted to a grid coordinate. Training uses cross-entropy (CE) and mean squared error (MSE) loss, reporting top-1 accuracy and L2 distance to the ground-truth position. The total loss is CE + 0.5× MSE, where MSE helps refine spatial predictions. Optimization is done via Adam (Kingma & Ba, 2015) with a cosine learning rate schedule, running for up to 30 epochs with early stopping (patience = 3) based on validation top-1 accuracy.

- **Patch Reconstruction.** A linear layer reconstructs the image patch from token embeddings ($D$), outputting $P^2 \times 3$ values (flattened RGB patch). Training minimizes MSE loss with AdamW (Loshchilov & Hutter, 2019), using a cosine learning rate schedule for up to 30 epochs, with early stopping (patience = 3) based on validation MSE.

- **Image Classification.** A linear classifier is trained on ImageNet-1K using 200,000 images containing at least one outlier token. Per image, a `[CLS]`, random normal, or random outlier token is selected. Training uses cross-entropy loss, Adam, a cosine learning rate schedule, and early stopping (patience = 3) based on validation accuracy.

**Claim 5.** We evaluate pretrained DINOv2 models - trained with and without registers - alongside the pretrained 1-layer classifier heads from the official DINOv2 repository [5]. These models are assessed on the ImageNet-1K validation set, with performance reported as top-1 accuracy. Additionally, we investigate the use of registers for classification, by training three linear models on pretrained DINOv2 architectures (small, base, and large), each incorporating four registers. The models leverage distinct image representations:

- **CLS + Patch Mean:** Concatenation of the `[CLS]` token and the mean of the output patch tokens.

---

[4]The sampled subset is available in our GitHub repository
[5]DINOv2 Repository

- **CLS + Patch Mean + Registers:** Concatenation of the `[CLS]` token, the mean of the output patch tokens, and all registers.

- **Patch Mean + Registers:** Concatenation of the mean of the output patch tokens and all registers.

These linear models are trained for 20 epochs using SGD with a cosine learning rate scheduler, and we report top-1 accuracy on the validation set.

### 3.4 Models

The training schemes of DINO, DINOv2, DeiT3, and OpenCLIP reflect distinct approaches tailored to their objectives. These models were selected to align with those used in the work of Darcet et al. (2024), ensuring consistency and comparability with prior work. DINO (Caron et al., 2021) and DINOv2 (Oquab et al., 2023) leverage a self-supervised teacher–student framework in which the teacher generates pseudo-labels from augmented image views using a momentum encoder and the student aligns to these via a cross-entropy loss. Both employ multi-crop augmentation and contrastive learning without negative samples, preventing collapse through centering and sharpening. DINOv2 further improves scalability and robustness for richer, more generalizable features and is the only model in the set that integrates the iBOT loss, which encourages tokens to retain local information by predicting the content of masked patches.

In contrast, DeiT3 (Touvron et al., 2022) follows a supervised paradigm, optimizing performance with extended training schedules, cosine annealing with warm restarts, and advanced augmentations such as RandAugment (Cubuk et al., 2020), Mixup (Zhang et al., 2018), and CutMix (Yun et al., 2019). OpenCLIP (Gadre et al., 2023) adopts a multimodal contrastive objective, jointly training Transformer-based image (e.g., ViT (Dosovitskiy et al., 2021)) and text (e.g., BERT (Devlin et al., 2019)) encoders on large-scale image–text pairs (e.g., LAION-142M). This yields strong zero-shot transfer at the cost of substantial computational resources due to its dual-encoder architecture and dataset scale.

Pyramid Vision Transformer v2 (PVT-v2) (Wang et al., 2022a) employs a hierarchical architecture optimized for dense prediction tasks. It incorporates spatial-reduction attention, which progressively downsamples key and value feature maps across pyramid stages to reduce computational overhead while preserving multi-scale representation capacity. Swin Transformer v2 (Swinv2) (Liu et al., 2021) extends its predecessor with shifted-window attention, balancing local and global context through window partitioning and cyclic shifting. Both PVTv2 and Swinv2 do not rely on a dedicated `[CLS]` token; instead, we follow their official implementations and obtain image-level representations via global average pooling over the final patch tokens.

### 3.5 Datasets

This section introduces the datasets used in our experiments: LVD-142M, LAION-2B, and ImageNet-1K. To begin with, LVD-142M (Oquab et al., 2023), a large-scale dataset with 142 million images, is used to train DINO and DINOv2, providing extensive visual data for robust feature learning. Moreover, LAION-2B (Schuhmann et al., 2022), utilized to train OpenCLIP, is distinguished by its vast scale and diversity, enabling strong image-text alignment and supporting tasks such as zero-shot classification and retrieval. Furthermore, ImageNet-1K (Deng et al., 2009), a widely used benchmark with over 1.2 million labeled images across 1,000 classes, is employed for training DeiT3, Swinv2, and PVTv2. Notably, Darcet et al. (2024) re-trained DINOv2 and DeiT3 on ImageNet-22K instead of using the mentioned datasets. Whether they re-trained OpenCLIP on ImageNet-22K or LAION-2B or used DataComp (Gadre et al., 2023) remains unspecified.

### 3.6 Hardware & $CO_2$

All experiments were conducted on the Snellius supercomputing cluster. Each node features 64 CPU cores and 4 NVIDIA H100 GPUs; we utilized 1/4 of a node (16 CPU cores + 1 GPU). In total, 30,000 Service Units (SBUs) were consumed, where 192 SBUs correspond to 1 hour of compute time on 1 H100 GPU and 16 CPU cores.

**$CO_2$ Emissions.** We estimated $CO_2$ emissions using:

$$CO_2 = CI \cdot PUE \cdot P \cdot t \tag{1}$$

where $CI$ is the Carbon Intensity (0.37 kg/kWh for Amsterdam Data Tower)[6], $PUE$ is the Power Usage Effectiveness (1.19)[7], $P$ is the power consumption (0.805 kW: 0.105 kW for CPU + 0.7 kW for GPU), and $t$ is the compute time (260.42 hours for 50,000 SBUs). The estimated emissions are 92.30 kg, highlighting the environmental impact of computational research and the need for energy-efficient practices.

## 4 Results & Analysis

In this section, we (i) reproduce the findings of Darcet et al. (2024)—extending their evaluation to additional Vision Transformers—and (ii) report further experiments with hierarchical-architecture models.

### 4.1 Reproduction & Core ViT Experiments

In this section, we focus our analyses on the models examined by Darcet et al. (2024). We examine each claim from Section 2 sequentially, cross-checking the reproduced metrics with the original paper. Then, we broaden the benchmark to cover an expanded set of models.

**Claim 1** We begin by clarifying a key distinction: Darcet et al. (2024) use "outlier," "artifact," and "high-norm token" interchangeably, assuming artifacts are high-norm outlier tokens. However, as detailed in the Experimental Setup, these terms are not always equivalent. Artifacts can appear in both attention maps and feature maps, but only in the latter can they be definitively linked to high norms.

The conflation in Darcet et al. (2024) arises from their exclusive use of DINOv2-G, where attention and feature maps are nearly identical. However, this does not generalize. We observe cases where artifacts appear in attention maps without corresponding to high-norm tokens in feature maps, indicating they are not true outliers. An example of this effect can be observed in Figure 2. A token is a high-norm outlier only if it appears exaggerated in the feature map. For example, in DINOv2-L, a dominant artifact in the attention map does not translate to a high-norm feature representation. Instead, it should be classified as a high-attention token, not a high-norm token.

Hence, while Claim 1 holds for DINOv2-G—the only model analyzed by Darcet et al. (2024)—it does not generalize. Our findings emphasize the need to differentiate artifacts, outliers, and high-norm tokens, as their equivalence depends on the model architecture and is not universal.

**Claim 2** Claim 2 states that training large models (>300M parameters) leads to high-norm token emergence. Figure 1 confirms this, showing artifacts in the feature maps of DINOv2-G, DeiT3-H, and OpenCLIP-H/G. Notably, these artifacts appear consistent across large models.

However, high-norm tokens are not exclusive to large models. Figure 1 also shows that smaller models, including DeiT3-S and OpenCLIP-B also exhibit outliers. The only exceptions are DINO, aligning with the observations of Darcet et al. (2024), and DINOv2-B. While outliers are prevalent in large models, our results reveal their frequent occurrence in smaller models as well. Figure 4 shows that OpenCLIP-B, a smaller model, has a comparable or even greater number of high-norm tokens than OpenCLIP-H, a larger counterpart. This suggests that model size alone does not determine outlier formation.

In summary, we confirm the findings of Darcet et al. (2024) on outliers in large models and extend them to smaller models (<300M parameters). Our results indicate that high-norm token emergence is not solely a function of model size, refining the original claim.

**Claim 3** In Figure 3, we reproduce the results of Darcet et al. (2024), confirming that high-norm tokens in DINOv2-G predominantly appear in areas with redundant local information. We plot the cosine similarity of

---

[6]www.nowtricity.com
[7]https://www.clouvider.com/amsterdam-data-tower-datacentre/

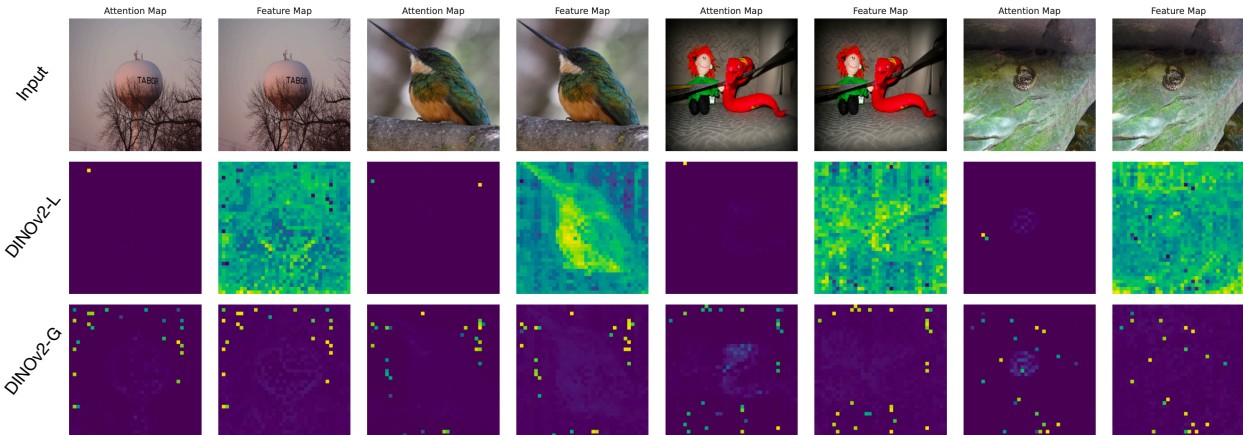

Figure 2: Attention maps and feature maps generated by DINOv2-L (second row), and DINOv2-G (last row) model variants, for four sample images. This example illustrates the difference between attention maps and feature maps. We observe that an image can have artifacts in the attention map and simultaneously have a much clearer feature map.

each patch with its four nearest neighbors, generating distributions for artifact and normal patches. Artifact patches are approximately 2.5 times more likely than normal patches to reside in low-information areas, aligning with the authors' conclusions.

Besides DINOv2-G we also extend the analysis to three additional models. The cosine similarity plots for these models closely resemble those of DINOv2-G - with the exception of DINOv2-S - validating the generalizability of the authors' claim. Notably, DINOv2-S, which lacks artifacts in its feature maps, exhibits an inverted pattern, i.e. high-norm tokens do not appear in regions of redundant visual information.

In conclusion, our experiments confirm the reproducibility of the findings of Darcet et al. (2024) and generalize them across a broader set of models, reinforcing the relationship between high-norm tokens and redundant local information.

**Claim 4** The experimental results, summarized in Table 1 and Table 2 below (detailed explanation in methodology in Section 3.3), demonstrate two key findings. First, our replication experiments with DINOv2-G align with the original findings reported by the authors: normal tokens exhibit superior performance to outlier tokens in both pixel reconstruction and position prediction tasks in terms of L2 error and Top-1 accuracy respectively. This supports the hypothesis that high-norm "outlier" tokens retain significantly less local information compared to normal tokens. Secondly, this pattern persists in smaller architectures (DeiT3-M, OpenCLIP-B, and DINOv2-S, which have less than 300M parameters), where outlier tokens similarly underperform normal tokens across tasks. Despite their reduced parameter counts, these models mirror DINOv2-G's token behavior, further corroborating the proposed relationship between token norm and local information content.

In addition, an important observation is that DINOv2-G, as well as DINOv2-S, demonstrate significantly better performance in comparison to the other models. This is due to the iBOT loss (Zhou et al., 2022) that models of the DINOv2 family implement. iBOT trains models to reconstruct masked patches and enhances their ability to capture and retain local structures and spatial relationships. This is highly effective for tasks that require understanding local image content and explains the superior performance of DINOv2 models in pixel reconstruction and position prediction tasks.

Moreover, for the second part, regarding the outlier tokens holding global information, we reproduce the experiments of Darcet et al. (2024), who trained a linear classifier using randomly sampled normal and outlier patches from DINOv2-G. Tokens are sampled from images, identified as outliers or normal patches, and used in place of the `[CLS]` token. A balanced dataset is created, and if high-norm outliers encode global information, their classification accuracy should surpass that of normal patches.

| Model | Top-1 Acc (%) ↑ | | Avg. Distance ↓ | | Baseline |
|---|---|---|---|---|---|
| | Normal | Outlier | Normal | Outlier | |
| DINOv2-S | **28.05** | 19.18 | **1.07** | 1.27 | 14.15 |
| DeiT3-M | **7.53** | 7.05 | 4.29 | **3.88** | 5.35 |
| OpenCLIP-B | 6.76 | **8.19** | **3.87** | 6.38 | 5.35 |
| DINOv2-G | **45.48** | 7.60 | **0.67** | 7.41 | 14.15 |
| Swin-S | **19.44** | 13.38 | 2.67 | **2.59** | 3.04 |
| PVT-b2 | 37.84 | **100.00** | 1.21 | **0** | 2.65 |

Table 1: Results for the position prediction task. Bold numbers indicate superior performance. We prioritize top-1 accuracy over average L2 distance, as the latter is susceptible to trivial solutions (e.g., always predicting the image center), especially under uniform distributions like ours. We report this trivial solution in the Baseline column for each model. Notably, for DINOv2-S, differences are minor, consistent with its lack of artifacts. OpenCLIP-B and particularly DINOv2-G support the findings of Darcet et al. (2024), where outlier tokens underperform normal tokens in position prediction. DeiT3-M and Swin-S exhibit contradictory results, with normal tokens showing higher top-1 accuracy despite worse average L2 distance. PVT-b2 achieves remarkable results, attaining 100% top-1 accuracy and 0 average distance.

| Model | L2 error ↓ | | Difference |
|---|---|---|---|
| | Normal | Outlier | Outlier − Normal |
| DINOv2-S | **383.13** | 521.56 | 138.43 |
| DeiT3-M | **801.07** | 1505.68 | 704.61 |
| OpenCLIP-B | **572.54** | 983.69 | 411.15 |
| DINOv2-G | **337.20** | 926.34 | 589.14 |
| Swin-S | 3945.00 | **3687.00** | -258.00 |
| PVT-b2 | **3826.00** | 5132.00 | 1306.00 |

Table 2: Results for the pixel reconstruction task. Bold numbers denote superior performance. Normal tokens generally exhibit a lower L2 error, except in Swin-S where outliers perform better. PVT-b2 and Swin-S show significantly larger values which show that pyramid architecture models do not perform well in pixel reconstruction tasks.

We extend this experiment to DINOv2-S, DeiT3-M, and OpenCLIP-B. DeiT3-M and OpenCLIP-B are included due to their similarity to DINOv2-G in cosine similarity patterns (see Figure 3), while DINOv2-S, which lacks high-norm tokens, serves as a contrasting case. We expect DINOv2-S to show minimal differences between patch types, further validating our hypothesis.

Results in Table 3 confirm the findings of Darcet et al. (2024) for DINOv2-G, with high-norm tokens outperforming normal patches. OpenCLIP-B and DeiT3-M exhibit similar patterns, with artifact-based accuracy exceeding normal patches by over 10%. In contrast, DINOv2-S shows negligible differences, as expected, given its lack of outlier tokens.

In conclusion, we confirm that high-norm tokens hold little local information and show that they also encode global information in OpenCLIP-B and DeiT3-M.

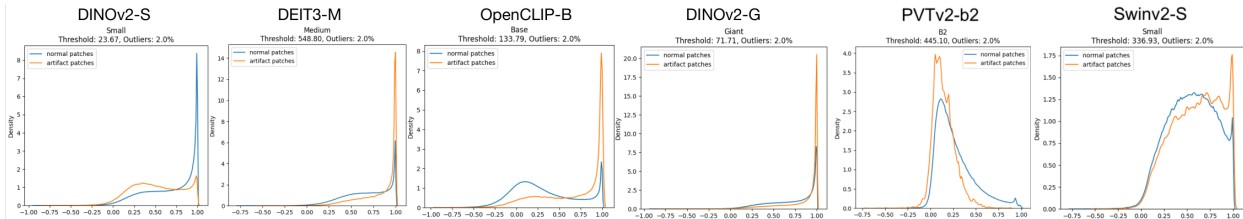

Figure 3: Cosine similarity plot of DINOv2-G with additional models. We include DeiT3-M and OpenCLIP-B, which exhibit similar behavior to DINOv2-G, and DINOv2-S, which lack artifacts, further validating the observed patterns. The figures demonstrate that cosine similarity is highest for normal patches in DINOv2-S, whereas in the other three models, outlier patches exhibit greater similarity. These results align with expectations, support the claims of Darcet et al. (2024), and extend them to additional models. Moreover, we also include Swin-S and PVTv2-b2. Both show an alternate behavior than the other ViTs, however, PVTv2-b2 is the only one that does not depict high cosine similarity for either normal or outlier patches.

| Category | DINOv2-S | DeiT3-M | OpenCLIP-B | DINOv2-G | PVT-b2 | Swin-S |
|---|---|---|---|---|---|---|
| [CLS] | **74.81%** | **81.08%** | **75.77%** | **81.56%** | **80.89%** | **82.01%** |
| Normal | 55.61% | 69.18% | 59.22% | 50.04% | 69.29% | 74.12% |
| Outlier | 57.00% | 78.96% | 69.48% | 68.33% | 63.54% | 76.16% |

Table 3: Image classification performance across all models. Bold numbers denote the highest score per model. Underlined values highlight the better of the two patch-based representations (Normal vs. Outlier).

**Claim 5** Darcet et al. (2024) address artifacts in attention maps by introducing registers. Rather than constraining the model, registers provide an alternative pathway for global information, reducing the reliance on redundant image tokens.

To validate this claim, we focus on DINOv2, as it is the only model with a pretrained version that includes registers. The results, depicted in Figure 5, clearly demonstrate that registers effectively eliminate artifacts from the attention maps. Our findings confirm the authors' claim and support the use of registers as a robust solution for mitigating outliers in attention maps.

Subsequently, the authors also claim that incorporating registers does not degrade classification performance and may occasionally improve it. To verify this, we evaluate pretrained DINOv2 models, with and without registers, on ImageNet-1K. As shown in Table 4, models with registers consistently perform marginally better, confirming the authors' claim. Notably, our DINOv2 model was trained on LVD-142M rather than ImageNet-22K, demonstrating the generalizability of their findings.

| Model | Regs × | Regs ✓ |
|---|---|---|
| Small | 81.15 | **81.65** |
| Base | 84.33 | **84.71** |
| Large | 86.12 | **86.80** |
| Giant | 86.90 | **87.29** |

Table 4: Reproduced ImageNet-1K classification results for the DINOv2 family. Bold numbers indicate superior top-1 accuracy (%). The results demonstrate that incorporating register tokens into the input layer does not degrade performance; instead, it consistently leads to a marginal improvement, confirming the results of Darcet et al. (2024).

| Representation | SMALL | BASE | LARGE |
|---|---|---|---|
| Pretrained Head | 81.65 | 84.71 | **86.80** |
| CLS+PM | **82.28** | **85.07** | 86.62 |
| CLS+PM+REG | 82.12 | 84.92 | 86.74 |
| PM+REG | 81.60 | 83.44 | 86.58 |

Table 5: Top-1 accuracy on the validation set for different image representations and DINOv2 model sizes. Bold numbers indicate superior performance. *PM* refers to patch mean, *reg* to register. The addition of register tokens does not appear to enhance classification performance, as the standard approach (CLS+PM) outperforms the alternatives that employ registers. The large model constitutes an exception, however the improvement is marginal and potentially due to stochasticity. Results using the pretrained head are also included for reference.

Furthermore, while the mechanism behind artifact generation remains unclear, the addition of registers effectively eliminates artifacts from attention maps, as shown in Figure 5.

In addition Darcet et al. (2024) hypothesize that register tokens - acting as replacements for outlier tokens - aggregate global information. To explore this, we retrain the classification head of a subset of models, leveraging register information for classification. We test three approaches: (1) concatenating registers with the [CLS] token and patch mean, (2) excluding the [CLS] token and using only registers and patch mean, and (3) concatenating the [CLS] token, patch mean, and registers. We also report the results of the pretrained head for reference. Results are presented in Table 5.

The table shows that adding registers does not improve performance. This outcome suggests that registers while accumulating global information, may not provide novel insights beyond the [CLS] token. Their contribution remains uncertain, as they are not explicitly trained to encode specific information, potentially limiting their utility for classification.

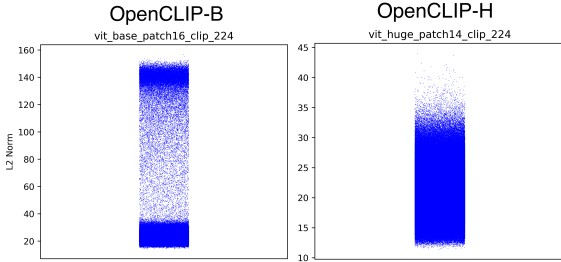

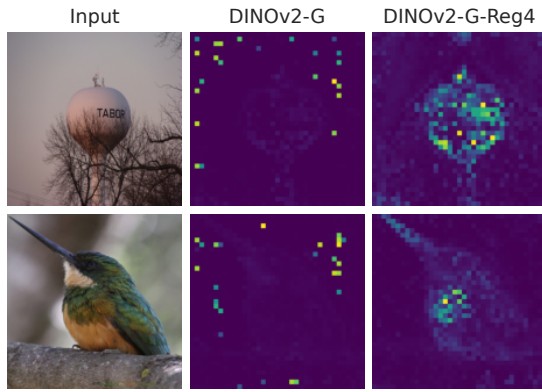

Figure 4: L2-norm distributions of OpenCLIP-H and OpenCLIP-B. Despite its smaller size, the base model exhibits significantly higher L2-norm values compared to the larger variant, indicating a greater susceptibility to artifacts. This further supports the observation that smaller models are affected by high-norm tokens. In some cases like this one, they can be more affected.

Figure 5: Attention maps generated by DINOv2-G without registers (left) and with registers (right). We observe that registers are instrumental in cleaning the attention map representations.

## 4.2 Hierarchical Vision Transformers

Next, we experiment further based on the results of Section 4.1 to see whether the claims of Darcet et al. (2024) apply to hierarchical vision transformers (H-ViTs), and gain more insight behind artifact generation.

Figure 6 depicts feature maps for PVT and Swin models. PVT exhibits high-norm tokens concentrated in localized regions, mirroring patterns in vanilla ViTs. The exception is PVT-b0, whose feature maps preserve spatial structures recognizable in the input image, akin to DINOv2-S (Figure 1). Swin, by contrast, displays high-norm tokens clustered in semantically coherent patches rather than isolated artifacts. These observations confirm that artifact emergence (claim 2) persists in hierarchical architectures, though their spatial distribution varies. Figure 7 further demonstrates bimodal L2-norm distributions in PVT models (except PVT-b0), with a distinct separation between normal and high-norm tokens. This motivated our selection of PVT-b2—exhibiting clear bimodality despite its modest size (25.4M parameters)—for downstream tasks.

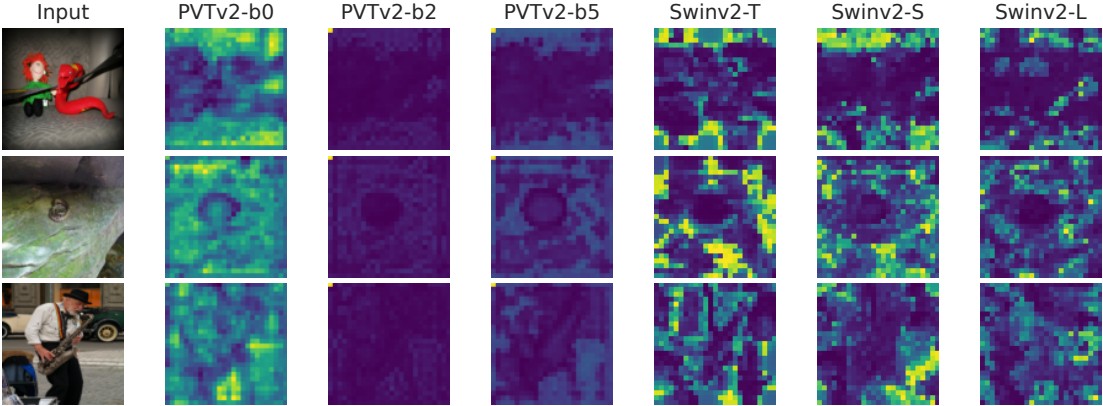

Figure 6: Feature maps generated by PVT and Swin models, calculated in high resolution. High-norm outliers are evident in PVT and Swin variants with different focused areas. All featured models are below 90M parameters, indicating that high-norm tokens also appear in small hierarchical vision transformers.

Figure 3 analyzes whether high-norm tokens in hierarchical ViTs emerge in low-information regions. Swin-S shows marginal similarity increases between artifact patches and neighbors compared to normal patches—substantially smaller than the difference in vanilla ViTs. PVT-b2 exhibits no meaningful distinction: both token types show near-zero cosine similarity, indicating orthogonal embeddings. These results

contrast sharply with vanilla ViTs, suggesting the relationship between token norms and local information (claim 3) is architecture-dependent. Hierarchical designs, particularly pyramid structures, may decouple token norm behavior from local redundancy through their multi-scale feature aggregation. The similarity of Swin-S with vanilla ViTs, although limited, motivated us to use it for the downstream tasks of claim 4. Subsequently, we will focus on the unique design of H-ViTs and explain the results illustrated in Tables 1–3.

**Swin-S** Normal-norm tokens encode markedly richer spatial cues than outliers (Top-1 position prediction = 19.44 % vs. 13.38 %). The pattern reverses for pixel reconstruction, where high-norm outliers yield a lower error (L2 = 3687 vs. 3945) and translate into a small advantage in image classification accuracy (76.16 % vs. 74.12 %). These observations suggest that Swin's windowed self-attention concentrates fine-grained appearance information in a small subset of tokens while distributing positional information more evenly across the feature map.

**PVT-B2** The pyramid variant exhibits the opposite split. Outlier tokens achieve perfect position prediction (100 % Top-1), yet underperform in pixel reconstruction (L2 = 5132 vs. 3826) and slightly in classification (63.54 % vs. 69.29 %). Because PVT progressively reduces spatial resolution, high-norm tokens may emerge primarily to preserve coarse spatial structure at each down-sampling stage, leaving texture details to the remaining tokens—consistent with PVT's strength on dense prediction tasks (Wang et al., 2022a). We analyze PVT's surprisingly high performance on the position prediction task in Appendix B.2.

**Implications** Taken together, Swin and PVT diverge from vanilla ViT behavior, underscoring that hierarchical attention reorganizes the division of labor between "normal" and "outlier" tokens. Whereas Swin shifts semantic richness to outliers, PVT assigns them positional control.

Our findings reveal that hierarchical ViTs exhibit distinct artifact behaviors compared to vanilla architectures. While high-norm tokens persist across model families (affirming claim 2's generality), their relationship to local information (claim 3) and downstream utility (claim 4) prove architecture-contingent. Pyramid structures like PVT decouple token norms from local redundancy, while shifted window mechanisms in Swin encourage semantically clustered outliers. These results qualify the original claims of Darcet et al. (2024), highlighting architectural considerations as critical moderators of token dynamics. Future artifact mitigation strategies must therefore account for structural inductive biases inherent to different ViT variants.

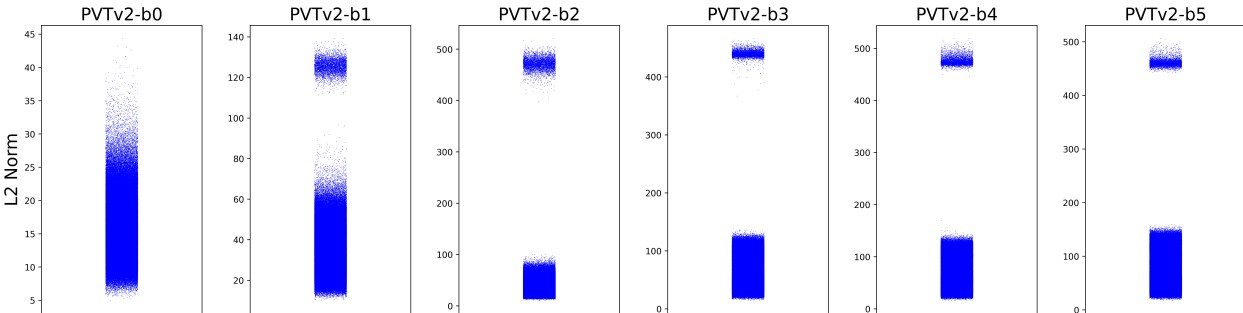

Figure 7: The L2 norms for all model sizes of PVT. The size of the model gets bigger from left to right. We notice that besides PVT-b0 all norm distributions are bimodal which is a sign of underlying artifacts.

## 5 Discussion

### 5.1 Reproducibility Work

In this work, we reproduce the findings of Darcet et al. (2024), confirming that large models (>300M parameters) are prone to outlier tokens in feature maps. Similarly to Darcet et al. (2024) we focus on DINOv2-G and demonstrate that high-norm tokens emerge in regions with redundant visual information. These tokens hold little local information, as evidenced by their poor performance in position prediction

and pixel reconstruction tasks, but encode global information, as shown by their superior performance in classification tasks using normal and outlier tokens. Overall, our results align closely with those of Darcet et al. (2024), successfully reproducing their key findings.

Finally, regarding the performance of registers, we conduct experiments for DINOv2-G and prove that adding register tokens in the input stage eliminates outlier tokens and leads to cleaner feature map representations. Registers significantly clean feature maps without degrading model performance, confirming their utility.

## 5.2 Generalization & Further Analysis

Beyond reproducibility, we generalize the findings of Darcet et al. (2024) by experimenting with additional models. As part of this generalization effort, we first address a key ambiguity in terminology that is highlighted when including more models in the experiments. We clarify the distinction between "artifacts" and "outliers," as well as "attention maps" and "feature maps," which are used interchangeably by Darcet et al. (2024) but are not equivalent in the general case. We note that the appearance of high-norm tokens in feature maps does not always align with the emergence of artifacts in attention maps.

Additionally, we explore the relationship between artifacts and model size, showing that smaller architectures with fewer than 300 million parameters (e.g., OpenCLIP-B and DeiT3-M) are also prone to outliers. However, this behavior is not universal, as models like DINOv2-S exhibit clear feature maps without artifacts.

Furthermore, we conduct a deeper investigation into the emergence of high-norm tokens in regions with redundant visual information and select a subset of models—OpenCLIP-B and DeiT3-M—that uniformly exhibit a significant presence of artifacts in their feature maps. By comparing outlier patches to normal ones using cosine similarity, we confirm that outliers in DeiT3-M and OpenCLIP-B are strongly associated with redundant regions, mirroring the behavior observed in DINOv2-G. In contrast, DINOv2-S, which lacks artifacts in its feature maps, shows no such association, as its outlier patches do not exhibit the same patterns.

To further assess the information content of outlier tokens, we perform position prediction, pixel reconstruction, and classification tasks for the same subset of models previously used. Outlier tokens perform poorly in position prediction and pixel reconstruction, confirming their lack of local information and aligning with the performance of DINOv2-G. However, they consistently outperform normal tokens in classification, indicating that they encode global information instead, which also aligns with the performance of DINOv2-G.

Moreover, acknowledging that registers hold global information, we attempt to utilize them in order to perform classification. However, their inclusion in classification tasks by concatenating them with the `[CLS]` token yields no improvements, suggesting that their global information may overlap with that of the `[CLS]` token. Their strong performance however is further proof that they do in fact hold global information.

Finally, extending the study to H-ViTs reveals a more heterogeneous landscape and underscores that the division of labor between normal- and high-norm "outlier" tokens is architecture-specific rather than universal. For Swin-S, outliers concentrate fine-grained appearance cues—yielding lower pixel-reconstruction error and a slight edge in classification—whereas normal tokens retain most spatial information. PVT-b2 exhibits the inverse pattern: outliers encode coarse positional structure (100% position-prediction accuracy) at the expense of texture reconstruction and classification, while normal tokens carry richer semantic content.

These opposing behaviors show that pyramid down-sampling, windowed attention, and other hierarchical design choices fundamentally reshape what information each token type stores. Thus, the global-information advantage of outlier tokens reported for vanilla ViTs does not generalize to H-ViTs. Any claim about token utility must therefore be qualified by the model's hierarchy, resolution schedule, and attention mechanism.

To sum up, while registers effectively address artifacts, the origin of artifact generation remains unclear. Our findings suggest that outlier tokens arise from the replacement of local information with global information in redundant regions, but the underlying cause of this behavior—and the source of increased L2 norms—remains an open question. This work highlights the need for further research into the mechanisms driving artifact generation in vision transformers.

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

## A    Feature Map Analysis

In this section, we conduct further analyses of high-norm tokens within the feature maps of ViTs. We extract and visualize the feature maps for the query, key, value, and the final feature map (prior to the final LayerNorm) to investigate their properties. The results are presented in Section A.1. Additionally, we perform a block-wise analysis of these feature maps, with findings detailed in Section A.2. Through these experiments, we aim to elucidate the emergence of artifacts and potentially pinpoint the origin of high-norm tokens.

### A.1    QKV Feature Maps

Our findings are illustrated in Figure 8. For the DINOv2-S model, the key feature maps exhibit high-norm tokens, while the same tokens manifest as low-norm outliers in the query and value feature maps. In the final feature map (prior to the last LayerNorm), most of these tokens no longer appear as high-norm outliers. Similarly, in DINOv2-L, specific tokens display distinct behavior depending on their location: they appear as high-norm outliers in the key feature maps but as low-norm outliers in the query and value feature maps. In the final feature map, only a very small subset of high-norm outliers persist, corresponding to those observed in the key feature maps. However, they are enough to completely harm the representation of the feature maps.

For DeiT-3-M, the query, key, and value feature maps contain a notable number of low-norm outliers. However, these do not translate into either high- or low-norm outliers in the final feature map. All feature maps in this model exhibit a noisy appearance. In contrast, OpenCLIP-B shows clusters of low-norm outliers across its query, key, and value feature maps. Intriguingly, in the final feature map, all tokens previously identified as low-norm outliers emerge as high-norm outliers.

From these observations, no consistent pattern of behavior emerges across the evaluated models. Further experimentation and analysis are required to better understand these phenomena and their implications. We leave further investigation of these findings to future work.

### A.2    Per-Block Feature Maps

To precisely determine the stage at which artifacts emerge in feature maps, we visualize the feature maps after each block during inference. Figure 9 reveals intriguing results. While artifacts appear in the early stages, the feature maps briefly improve before deteriorating again. Notably, in blocks 19-20, a single outlier in the K matrix in the upper left corner has increased the magnitude of the L2 norm in block 19, whereas this is not the case in the Q and V matrices. In the subsequent block, the norm of this outlier increases in the K matrix, while still having a low L2 norm in the Q and V matrices. However, rather than further degrading the visual representation of the feature map, the representation improves. This suggests that the underlying cause of poor feature map representations is either more complex than initially assumed or occurs within intermediate processing steps, possibly between layer normalization layers.

## B    PVT

PVT-v2 exhibited a very distinct behavior throughout all of our experiments. Therefore, in this section, we provide additional illustrations in order to shed some light on our results.

### B.1    Cosine Similarity

The prime motivation behind our selections of the specific model variants for the downstream tasks of claim 4 (position prediction, pixel reconstruction, image classification) was the cosine similarity between normal and artifact patches, depicted in Figure 3. However, PVT was the only model that showed a different behavior which led us to choose a different motivation for this case but also to partly reject the claim as it failed to be generalized. Since we plotted only PVT-b2 in the main paper, we believe it is interesting to see the rest

of the versions, which will also further support our conclusions. In Figure 10 we present the cosine similarity plot of every PVT variant.

## B.2 Position Prediction Results

In this section, we further investigate the surprisingly high accuracy of the PVT model in the position prediction task, as shown in Table 1.

### B.2.1 Architecture

The Pyramid Vision Transformer (PVT), particularly variants like PVTv2, departs significantly from the original Vision Transformer (ViT) architecture to better address dense prediction tasks. While ViT employs a columnar structure processing image patches at a single resolution, PVT adopts a hierarchical, multi-stage design reminiscent of Convolutional Neural Networks (CNNs).

A typical PVT architecture consists of four stages. Stage 1 processes the input image using an initial patch embedding layer. Notably, in PVTv2, this embedding is achieved via an overlapping patch embedding, which employs a 2D convolution with a stride smaller than its kernel size (e.g., $7 \times 7$ kernel, stride 4). This creates overlapping patches, enhancing local continuity compared to ViT's non-overlapping patching. Subsequent stages (2-4) progressively reduce spatial resolution while increasing channel dimensionality. Each stage transition utilizes similar overlapping patch embeddings (convolutional downsampling, e.g., $3 \times 3$ kernel, stride 2) followed by a series of Transformer blocks.

The architectural choices in PVTv2 introduce a strong spatial inductive bias, enabling the network to retain precise positional information, as evidenced by the perfect 100% top-1 accuracy observed when predicting the original grid location of output patch tokens. This sharply contrasts with standard ViTs, which typically show poorer performance on positional tasks. The key contributing factors are summarized below.

**Convolutional Embeddings and Downsampling**  The use of overlapping patch embeddings (strided 2D convolutions) for both the initial patch embedding and inter-stage downsampling inherently preserves spatial relationships. The overlapping nature ensures smooth spatial transitions, and the convolutional operation itself is spatially equivariant.

**Depthwise Convolution in MLP**  The inclusion of $3 \times 3$ depthwise convolutions (dwconv) within PVT blocks is critical. Unlike standard ViT MLPs that process each token embedding independently (relying solely on attention and positional encodings for spatial context), the dwconv explicitly models local spatial interactions between adjacent tokens on the feature grid within every block. This continuously reinforces spatial locality throughout the network's depth.

Collectively, these elements ensure that the final output tokens (e.g., the $7 \times 7$ grid) maintain a strong, traceable spatial correspondence to their original locations in the input image. The network's design prevents spatial information from being overly diffused or abstracted, leading to the observed high accuracy in position prediction tasks. This strong spatial bias makes PVT variants particularly effective backbones for dense prediction tasks where localization is paramount.

### B.2.2 Normal vs. Outlier Tokens

Given the strong spatial bias of PVT and its high accuracy on position prediction tasks, a natural question to ask is: *Why do outlier tokens outperform normal tokens?* To address this, we evaluate the trained position predictor with outliers separated at various thresholds, i.e., top 2%, 5%, and so forth. We plot the top-1 accuracy as a function of these thresholds and present the results in Figure 11. A clear trend is observed: tokens with higher L2 norms contain more positional information.

## C    Noise & Artifacts

In this section, we elaborate on the observed relationships between norm distributions and visual artifacts across a subset of our employed models. Those are also illustrated in Figure 12. We analyze each model depicted in the figure below:

**OpenCLIP-B** The norm distribution exhibits clear bimodality, characterized by a major cluster of tokens with low norms and a distinct minority of tokens displaying notably high norms. These high-norm outliers correspond directly to saturated areas, resulting in pronounced visual artifacts within the feature maps.

**DeiT3-S** The distribution is predominantly unimodal but skewed, with most token norms ranging between 200 and 900, accompanied by a significant tail extending towards higher norms (900–1400). This elongated tail contributes to scattered, high-norm tokens across the feature maps, manifesting as visual noise rather than clear artifacts.

**DINOv2-B** Tokens exhibit a symmetric, narrowly concentrated norm distribution with minimal tailing. Consequently, feature maps generated from DINOv2-B display negligible artifacts, indicating clean, stable visual representations.

This observed trend—transitioning from the relatively artifact-free DINOv2-B through the noisy DeiT3-S to the artifact-prone OpenCLIP-B—highlights that pronounced bimodality and distinct outlier clusters correlate strongly with significant visual artifacts, whereas distributions with skewed tails are associated with subtler noise.

## D    Feature Map Collection

In this section, we include the feature map of all models and their corresponding versions (Figure 13- 17).

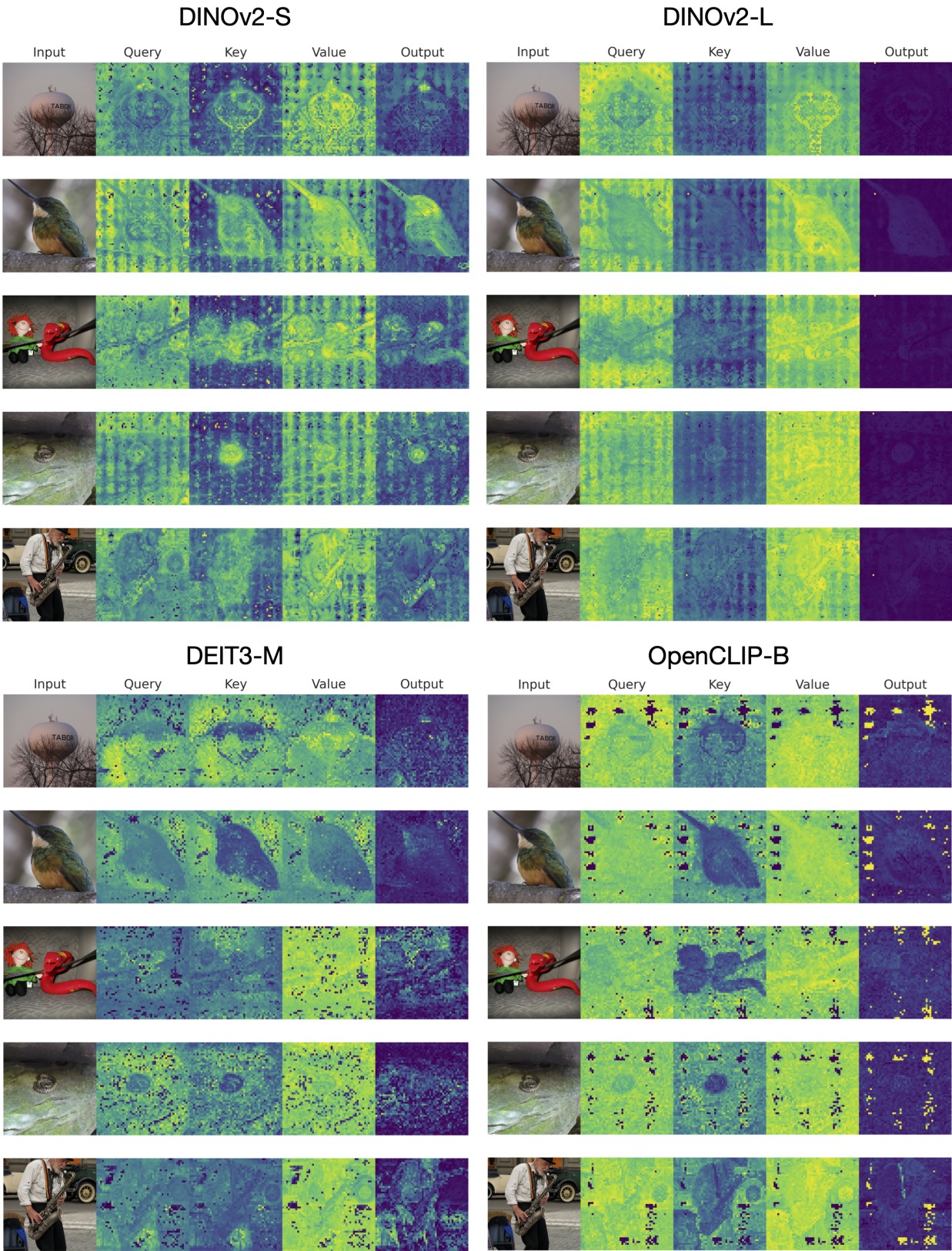

Figure 8: Feature maps generated by five sample images, for DINOv2-S, DINOv2-L, DeiT3-M, and OpenCLIP-B. We extract the tokens from the queries, keys, and value matrices of the last block, as well as the output tokens before the last LayerNorm of the last block. Computed in high resolution for better visualization.

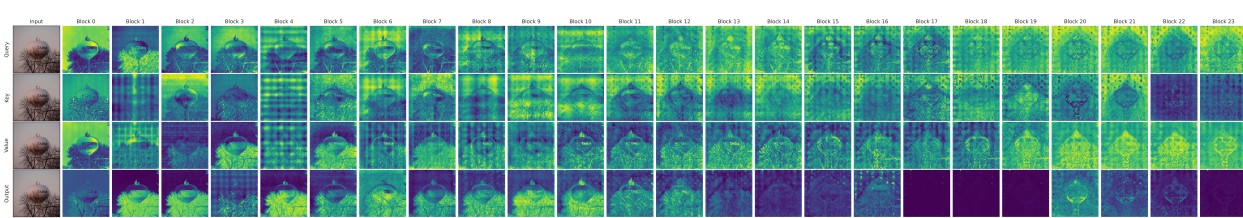

Figure 9: Feature maps generated per block layer for DINOv2-L. We observe that the resulting feature map representation starts to worsen after the mid-blocks. However, we notice that it starts to improve for a few blocks before being dominated again by artifacts.

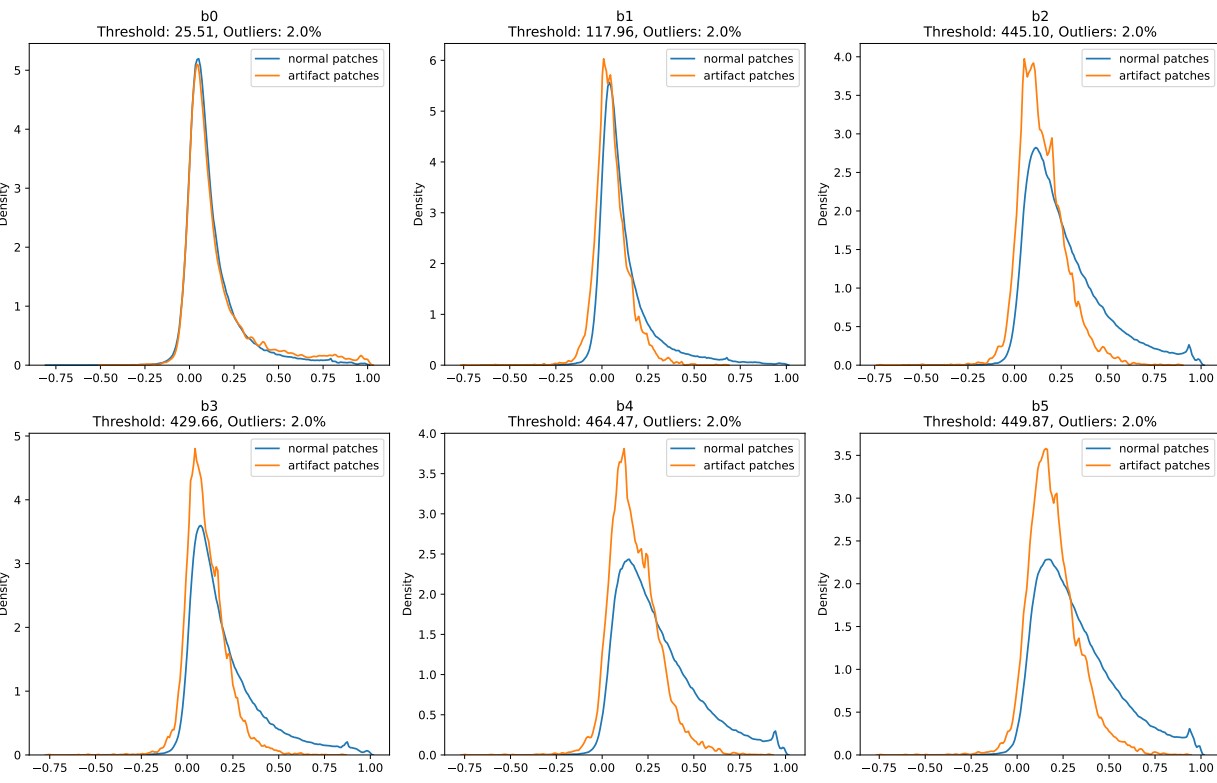

Figure 10: The cosine similarity plot of each PVTv2 variant. We see that they are all different from the ones seen in Figure 3 and therefore we choose an alternate motivation for our subsequent experiments.

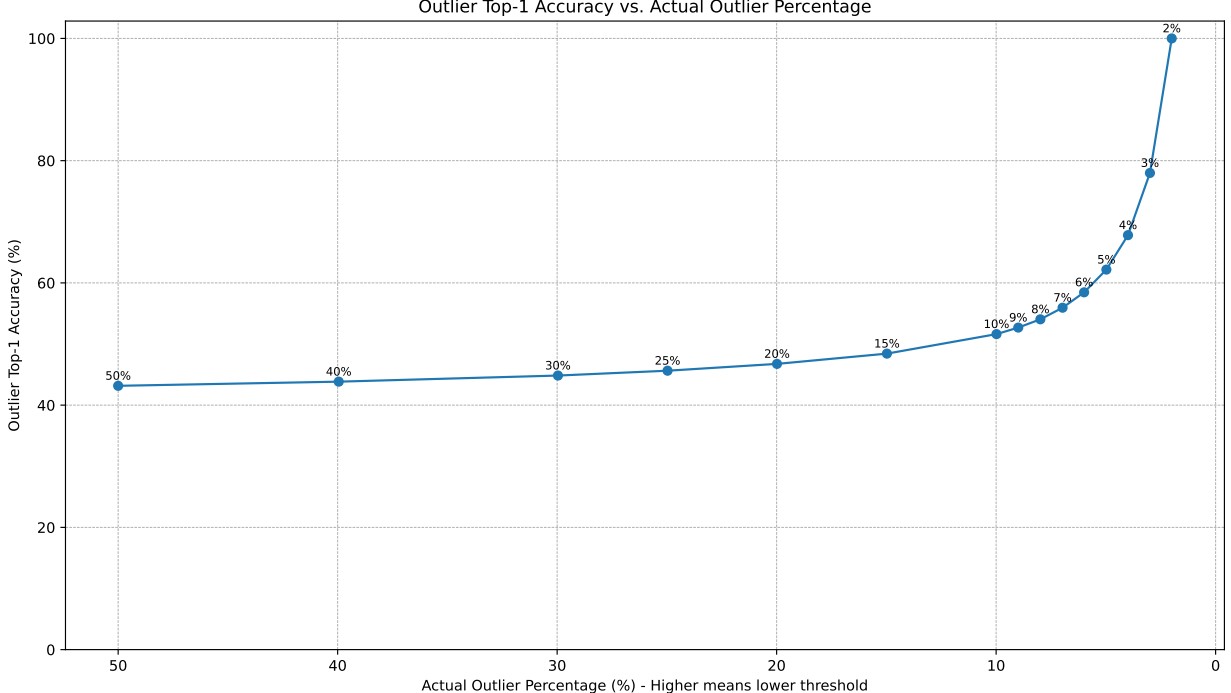

Figure 11: Trained position predictor evaluated using outliers at different thresholds. Outliers are designated progressively, starting from the top 50% tokens in terms of norm down to the top 2%. The top-1 accuracy for each evaluation is plotted, clearly showing that tokens with higher norms contain more positional information.

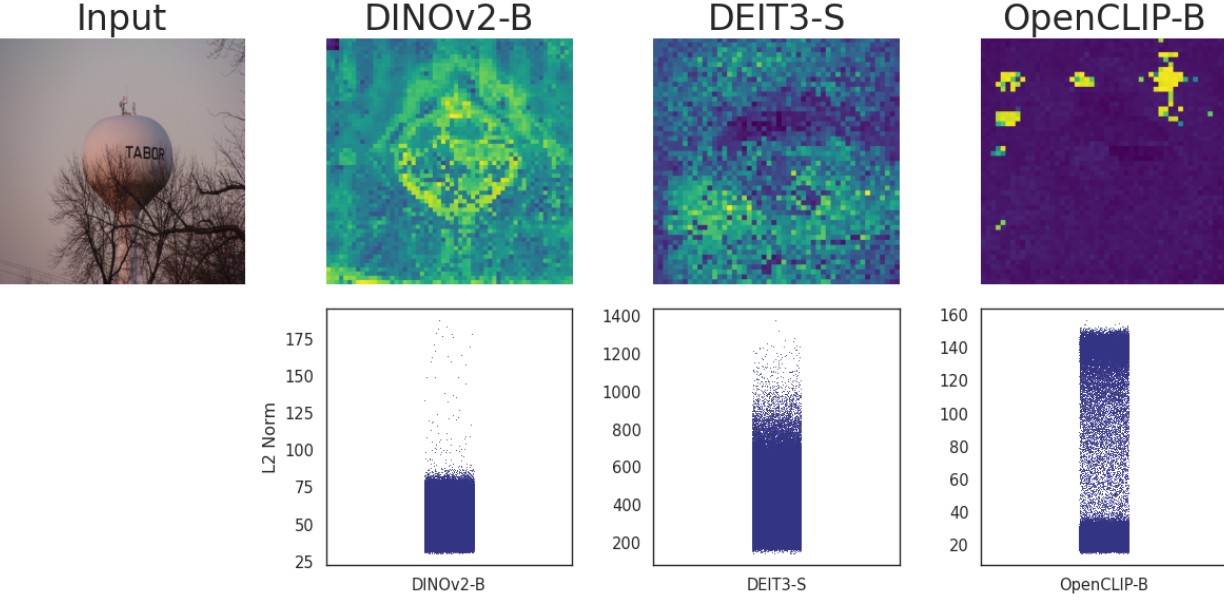

Figure 12: Top row: Feature maps of the input image generated by DINOv2-Base, DeiT3-Small, and OpenCLIP-Base models. Bottom row: L2 norm distributions of output tokens computed over 5,000 ImageNet-1k validation images. The feature map quality degrades as the norm distribution grows more pronouncedly long-tailed (increasing noise), with fully artifactual patterns emerging in cases of bimodal distributions, as observed in OpenCLIP-B.

Feature-Map Norms for DINOv2 Variants

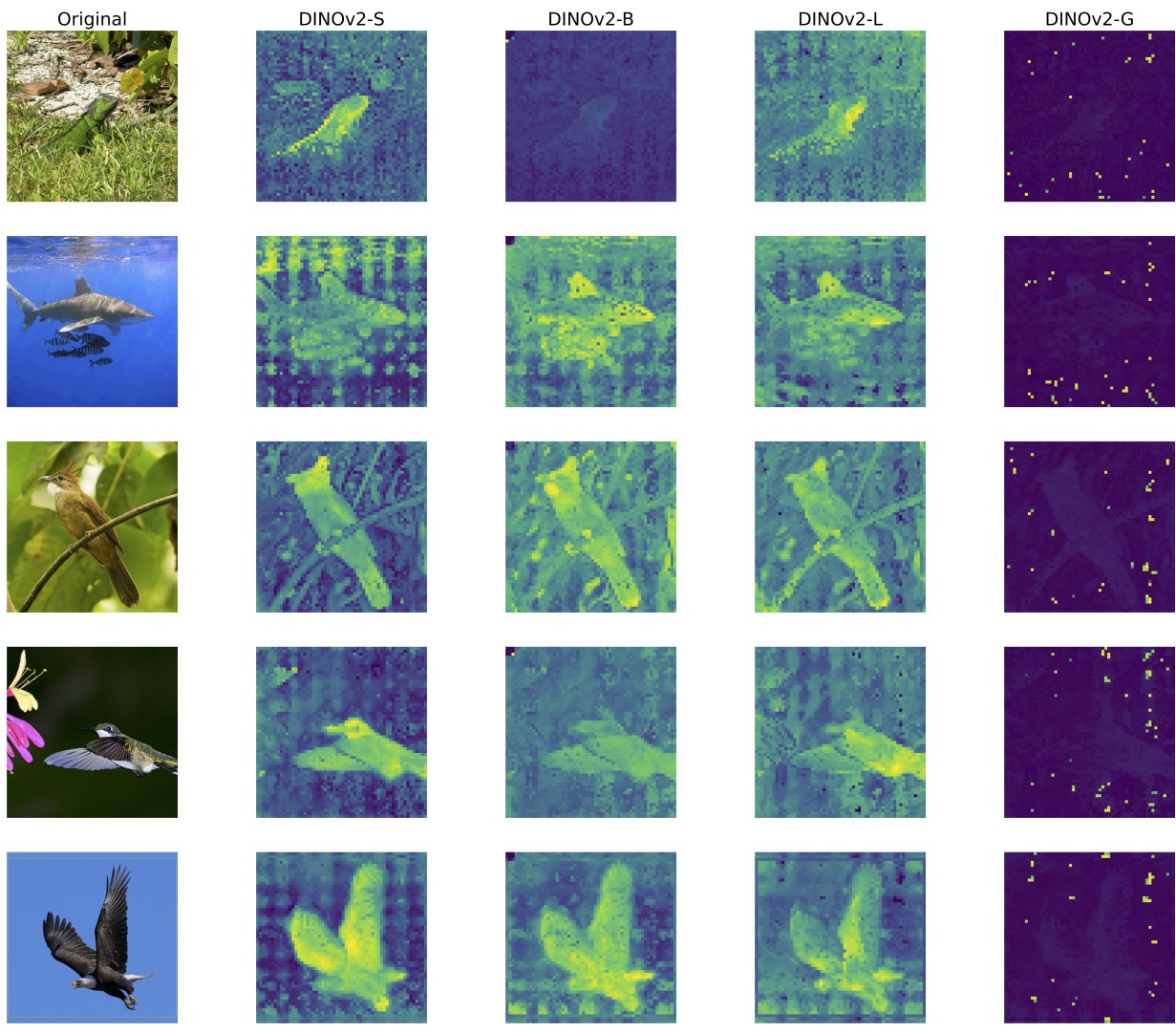

Figure 13: Feature maps for each DINOv2 variant.

Feature-Map Norms for OpenCLIP Variants

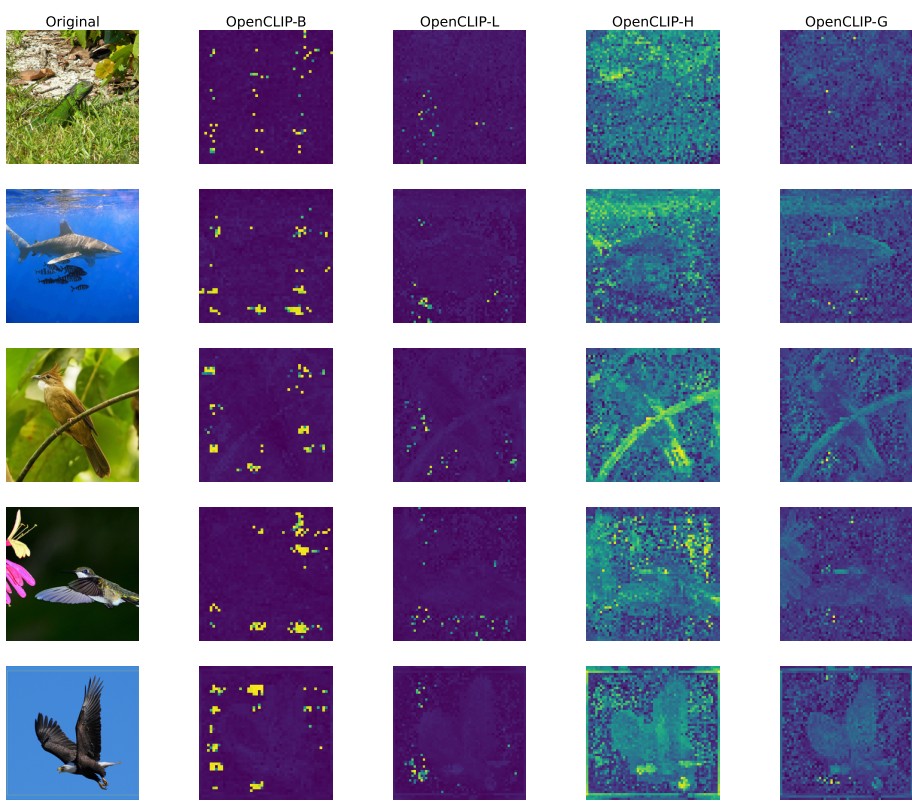

Figure 14: Feature maps for each OpenCLIP variant.

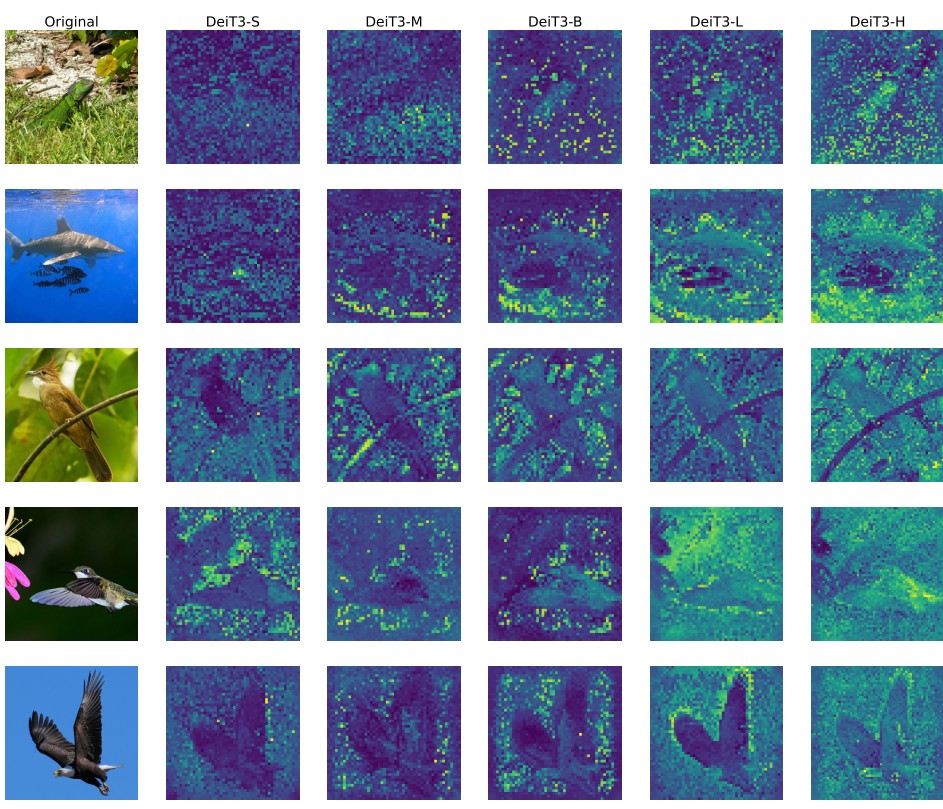

Figure 15: Feature maps for each DeiT3 variant.

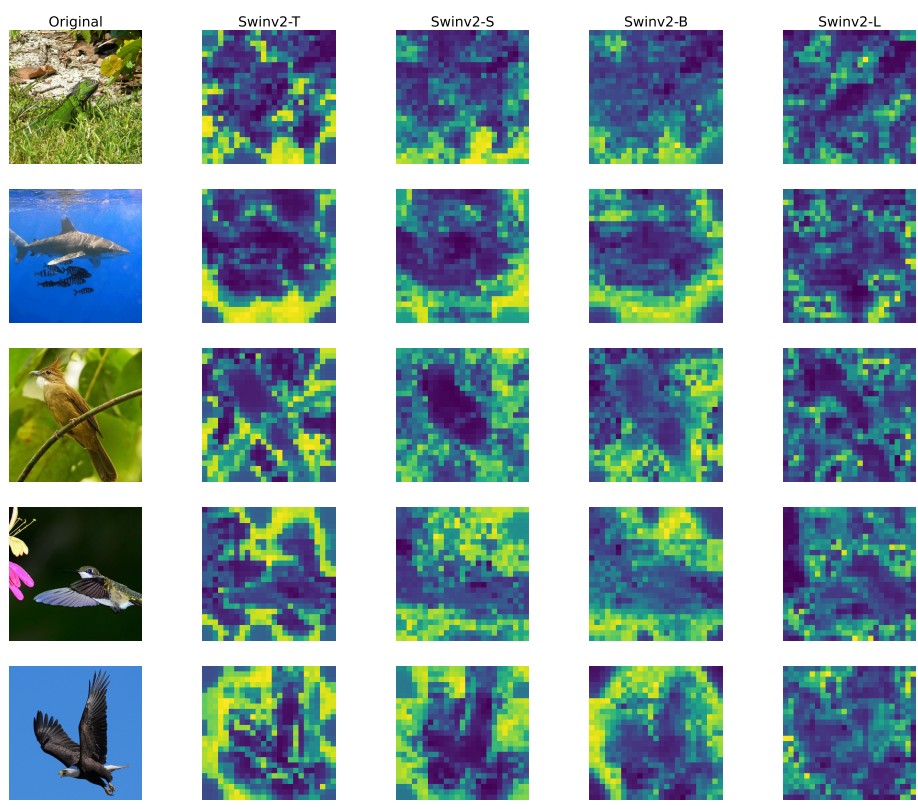

Figure 16: Feature maps for each Swinv2 variant.

Feature-Map Norms for PVTv2 Variants

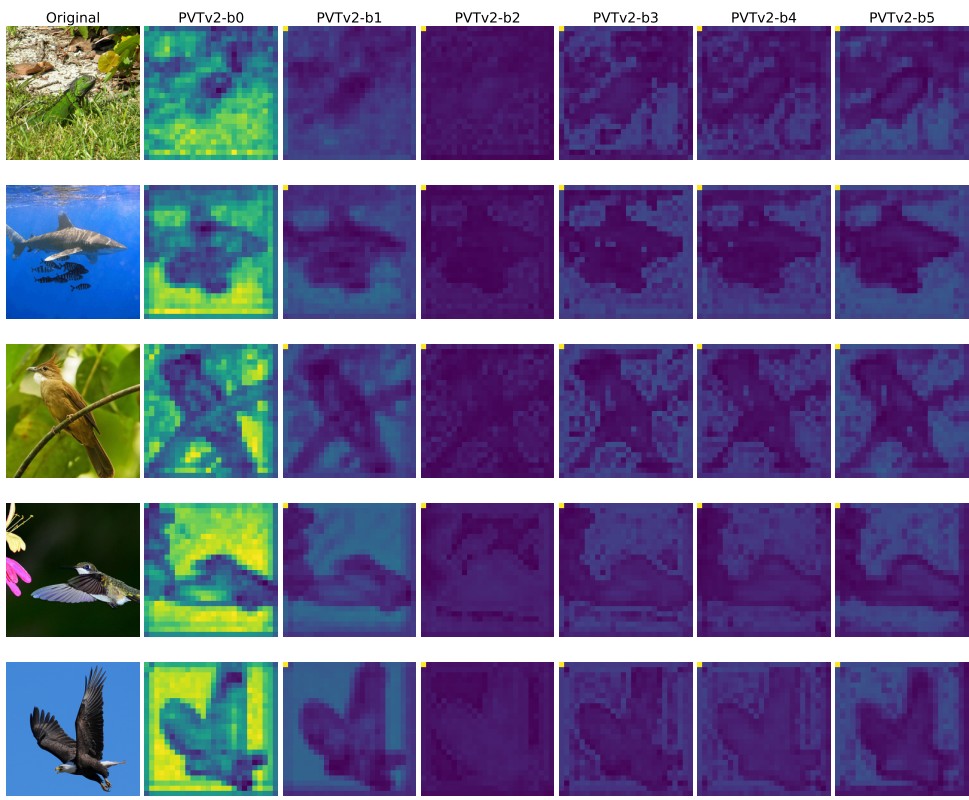

Figure 17: Feature maps for each PVTv2 variant.

