# OpenReview forum: "On the Reproducibility of Vision Transformers Need Registers"
_TMLR — Rejected by TMLR_

### Review · Reviewer_qVWm · 2025-04-03

**Summary Of Contributions:**

This paper reproduces the crucial findings of "Vision Transformers Need Registers" through extensive experiments and further extends the study by conducting additional experiments on smaller models, demonstrating the emergence of outlier tokens even in some small models.

**Audience:**

Yes

**Claims And Evidence:**

Yes

**Requested Changes:**

To strengthen the paper, I recommend that the authors conduct more experiments and provide a more in-depth analysis and discussion. Specifically, please address the weaknesses.

**Strengths And Weaknesses:**

**Strengths**

The reproduction process of this paper is highly detailed, with experimental designs strictly following previous works. The authors provide an exhaustive analysis, which can greatly facilitate readers' in-depth understanding of the attention artifacts and registers. The writing is also easy to follow.


**Weaknesses**
Although this paper may have some potential interest for the ML community, upon careful reviewing, the reviewer found that it provides limited new insights. Furthermore, attempting to experimentally verify the claims made in the previous paper fails to effectively provide readers with new knowledge.

The authors are suggested to provide more insightful findings. For instance, they could delve deeper into the underlying reasons behind the relationships between attention artifacts and model size/data size. Additionally, it is suggested that the authors conduct experiments on more vision models, particularly those with classic pyramid architectures, such as Swin and PVT. The reviewer is particularly interested in seeing whether the analysis can be generalized to these architectures. Furthermore, more visual tasks, such as semantic segmentation and object detection, should be considered, and the authors should explore how the design and performance of registers can be adapted to these tasks.

---

> ### Author Response · Authors · 2025-05-02
> **Reply to Reviewer qVWm**
>
> [Comment 1.1:]
> Although this paper may have some potential interest for the ML community, upon careful reviewing, the reviewer found that it provides limited new insights. Furthermore, attempting to experimentally verify the claims made in the previous paper fails to effectively provide readers with new knowledge.
> The authors are suggested to provide more insightful findings. For instance, they could delve deeper into the underlying reasons behind the relationships between attention artifacts and model size/data size.
>
> [Reply:]
> In this paper we initially tried to reproduce the results of Darcet et al. in an exhaustive manner, and subsequently provide additional generalizations and extensions to their work.
>
> As we attempt to clarify in our paper, we have generalized each and every one experiment of Darcet et al. by adding more models. The authors have used DeiT-3, DINO, DINOv2 and OpenCLIP in some small subset of their experiments, but in the majority of the experiments they use only DINOv2-G. From our side, each reproduced experiment involves at least 3 models that have been selected with specific motivation in each case to ensure whether the authors’ conclusions are generalizable.
>
> While we acknowledge that isolating the singular cause of artifact generation (e.g., interactions between model size, data complexity, or training dynamics) remains an open challenge, our work provides foundational empirical evidence of the problem’s scale and breadth.
>
> We agree with the reviewer that deeper mechanistic analysis (e.g., root causes tied to model/data size) would further enrich the field.
> However, we believe our study serves as a critical stepping stone by:
> 1. Validating the reproducibility of prior claims,
> 2. Highlighting the broader implications of artifacts across ViTs, and
> 3. Providing an empirical framework for future work to build upon.
> This area has significant grounds for future work and we hope our experiments can be a guide and inspire fellow researchers who will attempt to resolve this in the near future.
>
> [Comment 1.2:]
> Additionally, it is suggested that the authors conduct experiments on more vision models, particularly those with classic pyramid architectures, such as Swin and PVT. The reviewer is particularly interested in seeing whether the analysis can be generalized to these architectures.
>
> [Reply:]
> We thank the reviewer for this suggestion. To address it, we conducted new experiments incorporating Swin and PVT, two representative pyramid architectures. These models exhibit distinct behaviors in certain scenarios, which we analyze in detail in a dedicated section (Section 4.2). Their task-specific performance is now comprehensively documented in the revised Tables and Figures. We have also expanded the Discussion section to integrate insights from these findings, highlighting how pyramid architectures align with or diverge from trends observed in other models. For deeper exploration, Appendix B provides supplementary analysis of their unique characteristics.
>
> [Comment 1.3:]
> Furthermore, more visual tasks, such as semantic segmentation and object detection, should be considered, and the authors should explore how the design and performance of registers can be adapted to these tasks.
>
> [Reply:]
> The reason why we haven’t added any experiments regarding the performance of registers in tasks like object detection or semantic segmentation, is that we see our paper's contribution more of a deeper investigation on the appearance of artifacts rather than a focus on the solution of registers. We followed this path due to limited computing power, as DINOv2 remains the only model with available pretrained versions with registers, which limited the experiments that we could perform. We contacted the authors of Darcet et al. to gain access to more versions but unfortunately that was not possible. Thus, we are directly involved with registers only for the necessary reproduction part that confirms that for DINOv2 registers indeed assist in the removal of artifacts.

---

### Review · Reviewer_oDMp · 2025-04-10

**Summary Of Contributions:**

The authors try to reproduce claims in the paper Vision Transformers Need Registers. They verify five key claims of the paper. They confirm the validity of several of key claims, while also find that some claims do not extend universally to other models. They also explore the impact of model size to these claims.

**Audience:**

Yes

**Claims And Evidence:**

Yes

**Requested Changes:**

Add more observations and experiments about Register besides the 5 claims in the original Register paper. The extra claims should be significant and meaningful as the 5 original claims.

**Strengths And Weaknesses:**

Strengths:
- The experiments are conducted according to the five claims. Experiments are clearly presented.

Weaknesses:
- The experiments overall support the claims of the original Vision Transformers Need Registers paper. The extended conclusions, e.g., high-norm tokens are not exclusive to large models in claim 2, are not significant enough.
- No extra claims or observation of registers are proposed. The authors just verify the original conclusions proposed in the Register paper.

---

> ### Author Response · Authors · 2025-05-02
> **Reply to Reviewer oDMp**
>
> [Comment 2.1:]
> The experiments overall support the claims of the original Vision Transformers Need Registers paper. The extended conclusions, e.g., high-norm tokens are not exclusive to large models in claim 2, are not significant enough.
>
> [Reply:]
> Our study had two primary objectives: (1) reproducing the findings of Darcet et al., and (2) investigating whether their observations generalize beyond the original scope (e.g., to more ViT architectures, datasets, and model scales). While we acknowledge that Claim 2’s extension (high-norm tokens in smaller models) may appear incremental in isolation, it forms part of a broader novel finding: high-norm token behavior persists across architectures (columnar DeiT/DINO vs. hierarchical Swin/PVT), datasets, and model scales, suggesting these phenomena are not artifacts of specific design choices or sizes. We believe our study serves as a critical stepping stone by:
>
> 1. Validating the reproducibility of prior claims,
>
> 2. Highlighting the broader implications of artifacts across ViTs, and
>
> 3. Providing an empirical framework for future work to build upon.
>
> This area has significant grounds for future work and we hope our experiments can be a guide and inspire fellow researchers who will attempt to resolve this in the near future.
>
> [Comment 2.2:]
> No extra claims or observation of registers are proposed. The authors just verify the original conclusions proposed in the Register paper.
>
> Add more observations and experiments about Register besides the 5 claims in the original Register paper. The extra claims should be significant and meaningful as the 5 original claims.
>
> [Reply:]
> In order to be able to conduct experiments that would lead to concrete new claims about the contribution of registers and come up with direct comparisons with Darcet et al., we would have to have more pretrained models with registers or pretrain these models ourselves, which is computationally infeasible for us. Currently, the only publicly available pretrained version with registers is the one of DINOv2. We contacted the authors to gain access to more models but unfortunately that was not possible. Thus, we opted to focus on artifact generation and contribute by making a deeper investigation on the appearance of artifacts rather than a focus on the solution of registers.

---

### Review · Reviewer_J39V · 2025-04-15

**Summary Of Contributions:**

This manuscript details the reproduction of a prior work by Darcet et al. (2024) and the attempts at generalizing those findings to more models namely OpenCLIP and DeiT3.

The authors find that
1. Claims hold for DINO and DINOv2 which were studied in the original work [Darcet et al. (2024].
2. When extending to more models high norm tokens and artifacts in attention maps are no longer synonymous. One can happen without the other.
3. Several findings also hold for OpenCLIP and DeiT3.

**Audience:**

Yes

**Broader Impact Concerns:**

I do not see any broader impact concerns beyond those of the work of Darcet et al. (2024).

**Claims And Evidence:**

No

**Requested Changes:**

Discussed above in the weaknesses section.

**Strengths And Weaknesses:**

## Strengths

Reproducing this paper makes the results more reliable. This is of value to the TMLR community.

Alongside reproduction, new models have been tried and it appears that the findings do generalize in several cases.

Identifying the differences in findings when experimenting with OpenCLIP etc. further adds to the strengths of this manuscript.

## Weaknesses

**Results vs claims**: In several cases the evidence and the claims do not match exactly.
1. In figure 1 it is claimed that all models except DINO-S and DINOv2-B have high norm outliers. But only OpenCLIP-B and DINOv2-G have high norm outliers. DEIT3-S, DEIT3-H, OpenCLIP-H and OpenCLIP-G don't have any high norm outliers. The details are visible in the feature maps for these models. Am I missing something?
2. In Table 1, the story is less black and white than the text conveys. When looking at the avg. distance metric, which arguably is the more appropriate metric in this experiment, as expected DINOv2-S has negligible difference between normal and outlier tokens. DeiT3-M shows a reversed trend which counters the claims "normal tokens perform exceptionally well". The gap for OpenCLIP-B is also not as dramatic as expected. So for DeiT3-M one has to say that outlier tokens are still retaining positional information. This would also effect the line in page 11 ('perform poorly in position prediction').
3. In table 2, relative difference between normal and outlier L2 norm should be reported to make it easier for the reader.

**Missing details**:
1. In page 9, "Retraining the classification head" doesn't point to any new Table. Where could I find these numbers?

**Presentation**:
1. Section 3.2, 'a novel approach' overstates the novelty of the method used in Darcet et al. (2024). From Darcet et al. (2024), "This
mechanism was first proposed in Memory Transformers (Burtsev et al., 2020), improving translation tasks in NLP."
2. Please cite Adam and AdamW in page 4.
3. Could figure 2 be made into a floating figure to not take up the entire space? Alternatively one could transpose the figure and add a few more examples to fill the space.

---

> ### Author Response · Authors · 2025-05-02
> **Reply to Reviewer J39V**
>
> Dear Reviewer J39V,
>
> Given your multiple comments we will answer in two chunks due to the character limit.
>
> [Comment 3.1:]
> In figure 1 it is claimed that all models except DINO-S and DINOv2-B have high norm outliers. But only OpenCLIP-B and DINOv2-G have high norm outliers. DEIT3-S, DEIT3-H, OpenCLIP-H and OpenCLIP-G don't have any high norm outliers. The details are visible in the feature maps for these models. Am I missing something?
>
> [Reply:]
> Figure 1 depicts the feature maps of various models. The "ideal" feature map should be a representation that focuses on the most important objects of the image and to have limited to none noise. Similarly, the "ideal" attention maps should gather all the attention to the most important objects. These ideal maps can then be used for downstream tasks (e.g. object localization / detection / discovery).
>
> We see that DINO-S has a small amount of noise but besides that it has a very clean representation. DINOv2-B is also on the same category. However, the rest of the feature maps are very noisy and make it impossible to understand which are the important objects and where are they located in the image. OpenCLIP-B and DINOv2-G have concentrated on a small subset of pixels which makes their representation visibly worse.
>
> But, this does not mean that the rest of the models (namely DeiT3-S, DeiT3-H, OpenCLIP-H, OpenCLIP-G) do not display artifacts, as they are also heavily noised and it is difficult to understand what is happening on the image. In particular, both DeiT3 models introduce a lot of high-norm tokens in irrelevant areas that are not the central theme/objects of the image. Both OpenCLIP models show a similar, although less pronounced, behavior. In all depicted models, except DINO-S & DINOv2-B, artifacts appear either as few localized very high-norm tokens or as a higher number of still high-norm tokens, making the feature map unusable for downstream tasks due to noise.
>
> [Comment 3.2:]
> In Table 1, the story is less black and white than the text conveys. When looking at the avg. distance metric, which arguably is the more appropriate metric in this experiment, as expected DINOv2-S has negligible difference between normal and outlier tokens. DeiT3-M shows a reversed trend which counters the claims "normal tokens perform exceptionally well". The gap for OpenCLIP-B is also not as dramatic as expected. So for DeiT3-M one has to say that outlier tokens are still retaining positional information. This would also effect the line in page 11 ('perform poorly in position prediction').
>
> [Reply:]
> 1) With regards to the more appropriate metric for this task, we argue that avg. L2 distance is a worse metric than top-1 accuracy. In the L2‐distance metric, the optimal constant predictor is the geometric median of the target positions. For a uniform distribution, as is the case in our experiment, that’s the image center. The model can cheat this metric, since  “always predict center” minimizes the metric. In contrast, Top-1 accuracy requires an exact match to each position, making it immune to the trivial constant-prediction solution.
>
> 2) Regarding the reviewer's statement that "DINOv2-S has negligible difference between normal and outlier tokens". This is true in terms of the avg. L2 distance metric, but as we argued, top-1 accuracy is a harder and more reliable metric in this task. For this model, normal tokens clearly outperform outlier tokens in terms of top-1 accuracy. Although in models like DINOv2-G - which emphatically shows artifacts - this difference is more pronounced, the DINOv2-S results show that even when a model does not exhibit artifacts, normal tokens can still hold more local information than outlier tokens.
>
> 3) Then, regarding the statement "DeiT3-M shows a reversed trend which counters the claims normal tokens perform exceptionally well", it is true that DeiT3-M shows a reversed trend and our explanation is not clear enough. We have updated the text to be more elaborative and have removed the statement "normal tokens perform exceptionally well" which was not a good description in this scenario.
>
> 4) Subsequently, for the statement "The gap for OpenCLIP-B is also not as dramatic as expected.", we would like to disagree as the difference is almost double in the avg distance metric. It is not as important as in DINOv2-G where it is extremely big (which is why the authors of Darcet et al. might have chosen it), but we deem it is enough to support the claim.
>
> 5) Finally, for the statement "So for DeiT3-M one has to say that outlier tokens are still retaining positional information. This would also effect the line in page 11 (’perform poorly in position prediction’).", we have resolved this in our previous response.

---

> > ### Comment · Reviewer_J39V · 2025-05-05
> >
> > Reply to [Reply:] for [Comment 3.1:]
> > - As to which regions are salient depends on the task. In this case no task is specified so all background regions should also be highlighted in the feature maps. The central theme in the first figure could, for example, be the wavy lines on the ground which DEIT3-S identifies but DINO-S misses.
> > - "a higher number of still high-norm tokens": colour bars are necessary to support this. Are the norms really high relative to the other points. How does this vary between OpenCLIP-B (very clear outliers), DEIT3-S (noisey but not real outliers) and DINOv2-B (some noise but not real outliers).
> >
> > Reply to [Reply:] to [Comment 3.2:]:
> > - Agreed that the avg. L2 distance can be cheated, but Top-1 accuracy does not distinguish between being off by a bit vs being completely off. A higher top-1 accuracy but lower avg. L2 distance likely means that the position is roughly correct but not very exact. It is not sufficient evidence to conclude that position information is missing altogether. Could you report the accuracy of this center blind baseline? Do outliers perform comparably to this baseline?

---

> > > ### Author Response · Authors · 2025-05-10
> > > **Reply to Reviewer's J39V Additional Comments**
> > >
> > > We thank the reviewer for the additional feedback! Below, we will address once again each comment separately:
> > >
> > > [Comment 3.8:]
> > > As to which regions are salient depends on the task. In this case no task is specified so all background regions should also be highlighted in the feature maps. The central theme in the first figure could, for example, be the wavy lines on the ground which DEIT3-S identifies but DINO-S misses.
> > >
> > > [Reply:]
> > > To begin with, we want to note that: DeiT3, Swinv2, PVT-v2 are label-supervised models trained for classification on ImageNet-1k, DINO & DINOv2 are self-supervised models trained on LVD-142M for visual feature learning, and OpenCLIP is a text-supervised model trained on LAION-2B for text-image alignment.
> > >
> > > The reviewer accurately observes that DeiT3-S captures fine-grained patterns (e.g., wavy ground textures in Figure 1) missed by DINO-S. However, this specificity does not negate the broader issue: DeiT3-S exhibits substantially noisy feature maps that negatively impact downstream tasks. We posit that this noise comes from architectural artifacts, as evidenced by Darcet et al. (Table 3), who demonstrate that DeiT3 variants with registers achieve superior performance in object discovery (LOST benchmark).
> > >
> > > Unfortunately, we cannot directly reproduce these results, due to having no access to register-augmented models. However, considering our experiments and the rest of our results in the other tasks, we believe that the results of Table 3 and the corresponding conclusion that the authors reach are valid and align with our understanding.
> > >
> > > Finally, OpenCLIP (as you probably saw in Darcet et al.'s Table 3) has a distinct behavior that they analyze further in their Appendix C. We haven't focused our work into this open problem, but we see that OpenCLIP overall has similar trends with the other models in the majority of our experiments, whether that is feature map representations or a position prediction task. Thus, it is safe to assume that it is also facing the problem of artifact generation, even if there is another underlying factor/issue that hasn't been resolved by academia so far.
> > >
> > > [Comment 3.9:]
> > > "a higher number of still high-norm tokens": colour bars are necessary to support this. Are the norms really high relative to the other points. How does this vary between OpenCLIP-B (very clear outliers), DEIT3-S (noisey but not real outliers) and DINOv2-B (some noise but not real outliers).
> > >
> > > [Reply:]
> > > We have included Figure 12 in Appendix C which shows feature maps alongside norm distributions for DINOv2-B, DeiT3-S, and OpenCLIP-B, as suggested. The overall trend about the value of the high-norm tokens relative to the lower norm tokens can be seen using the y-axis labels.
> > >
> > > Below, we clarify the relationship between norm distributions and artifacts across models:
> > >
> > > OpenCLIP-B (clear artifacts): The norm distribution is bimodal, with a dominant cluster of low-norm tokens and a distinct minority of high-norm outliers. These outliers correspond to the saturated regions (artifacts) in the feature maps.
> > >
> > > DeiT3-S (very noisy): The distribution is unimodal but skewed, with the majority of tokens in [200, 900] and a long tail in [900, 1400]. While no bimodal distribution exists, the tail causes "noise" (scattered high-norm tokens) in feature maps.
> > >
> > > DINOv2-B (relatively clean): Norms follow a tight, symmetric distribution with minimal tail. Feature maps lack artifacts.
> > >
> > > This progression—from DINOv2-B (relatively clean) to DeiT3-S (noisy) to OpenCLIP-B (artifacted)—shows that bimodality (distinct outlier clusters) correlates with severe artifacts, while skewed tails correspond to subtler noise.
> > >
> > > [Comment 3.10:]
> > > Agreed that the avg. L2 distance can be cheated, but Top-1 accuracy does not distinguish between being off by a bit vs being completely off. A higher top-1 accuracy but lower avg. L2 distance likely means that the position is roughly correct but not very exact. It is not sufficient evidence to conclude that position information is missing altogether. Could you report the accuracy of this center blind baseline? Do outliers perform comparably to this baseline?
> > >
> > > [Reply]
> > > Although we agree with the reviewer that Top-1 accuracy does not distinguish between being off by a bit vs being completely off, we have included these two specific metrics since they were included in the original Darcet et al. paper, and our aim is in part to reproduce their findings. We believe that reporting Top-5 accuracy could potentially be a middle ground. However, we do not have the computing power at this point to do so, as it would require to re-run all of our experiments for position prediction.
> > > To remediate this, we have calculated and reported the baseline of the trivial solution (always predicting the middle) for each model, in Table 1.

---

> > > > ### Comment · Reviewer_J39V · 2025-05-16
> > > > **Reply**
> > > >
> > > > > "a higher number of still high-norm tokens": colour bars are necessary to support this.
> > > >
> > > > I meant a colourbar (https://matplotlib.org/stable/api/_as_gen/matplotlib.pyplot.colorbar.html) that accompanies the feature maps itself. But I guess this works as well or is complementary.
> > > >
> > > > Thank you for addressing these comments.

---

> ### Author Response · Authors · 2025-05-02
> **Reply to Reviewer J39V (2)**
>
> [Comment 3.3:]
> In table 2, relative difference between normal and outlier L2 norm should be reported to make it easier for the reader.
>
> [Reply:]
> This change has been integrated to the revised manuscript.
>
> [Comment 3.4:]
> In page 9, "Retraining the classification head" doesn't point to any new Table. Where could I find these numbers?
>
> [Reply:]
> We thank the reviewer for this comment. The mentioned statement was inaccurate and has been deleted. We indeed performed experiments that re-trained the classification head, however Table 4 contains the results with the pretrained heads only. The re-trained classification heads for DINOv2 versions with registers can be seen at Table 5 (notice that the values of DINOv2-L align between the two tables). However, there is no comparison regarding re-trained heads between DINOv2 versions with registers and without registers, as we deemed that it did not offer valuable information enough to justify the demanded compute.
>
> [Comment 3.5:]
> Section 3.2, 'a novel approach' overstates the novelty of the method used in Darcet et al. (2024). From Darcet et al. (2024), "This mechanism was first proposed in Memory Transformers (Burtsev et al., 2020), improving translation tasks in NLP."
>
> [Reply:]
> We thank the reviewer for this comment. Adding empty tokens to the input for storage purposes is not new to our field and it is incorrect to describe the method as novel.
>
> [Comment 3.6:]
> Please cite Adam and AdamW in page 4.
>
> [Reply:]
> We apologize for this oversight. We have now cited both Adam and AdamW in the revised manuscript.
>
> [Comment 3.7:]
> Could figure 2 be made into a floating figure to not take up the entire space? Alternatively one could transpose the figure and add a few more examples to fill the space.
>
> [Reply:]
> Figure 2 has been updated accordingly in the revised manuscript for better illustration purposes.

---

> > ### Comment · Reviewer_J39V · 2025-05-05
> >
> > Thank you for addressing these comments.

---

### Decision · Action_Editor_BpMg · 2025-05-31

**Recommendation:** Reject

**Comment:**

After revision, most of the minor comments are addressed, and the study is extended to hierarchical ViT models. However, the reviewers end up not being convinced.
The unresolved key concerns are as follows:

- While the scope of models and tasks was broadened, core concerns about “insufficient new insights” remain largely unaddressed (qVWm, oDMp).

- "... story is less black and white than the text conveys” (J39V): The authors made definite claims, but the reviewers found that the claims are not clearly true, which is deemed to be overstated.

In addition, the following suggestions are not addressed.

- “more visual tasks, such as semantic segmentation and object detection” (qVWm)
- “Top-1 accuracy does not distinguish between being off by a bit vs being completely off” and request for “accuracy of this center blind baseline” (J39V)



------------------------------------------------------

Note: This work [2] is largely overlapped with the concurrent submission [1].

[1] Reproducibility Study of “Vision Transformers Need Registers” https://openreview.net/forum?id=w9pgM58H05
(submitted on 25 Feb 2025 (modified: 27 Feb 2025))

[2] On the Reproducibility of Vision Transformers Need Registers https://openreview.net/forum?id=4VonR2EPrf
(submitted on 01 Mar 2025 (modified: 12 Mar 2025))

**Audience:**

This work shares a similar audience with that of [Darcet et al. (2024)], but the coverage is smaller because no experiments were added to clarify the effects of the claims in vision downstream tasks.

**Claims And Evidence:**

This manuscript “reproduces a notable prior work [Darcet et al. (2024)] successfully and generalizes the findings to more models” (Reviewer J39V), showing that “when extending to more models high norm tokens and artifacts in attention maps are no longer synonymous” (J39V). It further “verifies five key claims of the paper” across both self-supervised and supervised Vision Transformers and “extends the study by conducting additional experiments on smaller models” (qVWm). New experiments on pyramid architectures (Swin, PVT) assess whether these phenomena hold beyond the original ViT designs.

The reviewers acknowledge the exhaustive analysis and clearly presentation. However, Reviewer J39V concerns that
“Top-1 accuracy does not distinguish between being off by a bit vs being completely off,” and requests for “accuracy of this center blind baseline,” but not addressed by the authors.

Given the reviewers' concerns on alignment of the claims and evidence, carefully selecting additional metrics would have been beneficial to support stronger or clearer conclusions. It is understandable that the work follows the same setting as the original work [Darcet et al. (2024)]. However, since the claims of this manuscript are shifted from the original work, the authors may design a better metric to sharpen the claims of the submission.

Also, another main claim is the relationship with the smaller models, but this is deemed to be not significant enough (qVWm, oDMp).

**Resubmission Of Major Revision:**

The authors may consider submitting a major revision at a later time.